# Massive endocytosis triggered by surface membrane palmitoylation under mitochondrial control in BHK fibroblasts

Donald W Hilgemann[1]*, Michael Fine[1], Maurine E Linder[2], Benjamin C Jennings[2], Mei-Jung Lin[1]

[1]Department of Physiology, University of Texas Southwestern Medical Center, Dallas, United States; [2]Department of Molecular Medicine, Cornell University College of Veterinary Medicine, Ithaca, United States

**Abstract** Large Ca transients cause massive endocytosis (MEND) in BHK fibroblasts by nonclassical mechanisms. We present evidence that MEND depends on mitochondrial permeability transition pore (PTP) openings, followed by coenzyme A (CoA) release, acyl CoA synthesis, and membrane protein palmitoylation. MEND is blocked by inhibiting mitochondrial Ca uptake or PTP openings, depleting fatty acids, blocking acyl CoA synthesis, metabolizing CoA, or inhibiting palmitoylation. It is triggered by depolarizing mitochondria or promoting PTP openings. After mitochondrial MEND blockade, MEND is restored by cytoplasmic acyl CoA or CoA. MEND is blocked by siRNA knockdown of the plasmalemmal acyl transferase, DHHC5. When acyl CoA is abundant, transient $H_2O_2$ oxidative stress or PKC activation initiates MEND, but the immediate presence of $H_2O_2$ prevents MEND. The PTP inhibitor, NIM811, significantly increases plasmalemma in normally growing cells. Thus, the MEND pathway may contribute to constitutive as well as pathological plasmalemma turnover in dependence on mitochondrial stress signaling.

*For correspondence: donald.hilgemann@utsouthwestern.edu

Competing interests: The authors declare that no competing interests exist.

## Introduction

Endocytic processes that do not use classical endocytic proteins remain poorly understood because they are mechanistically inhomogeneous and difficult to study in isolation (*Mayor and Pagano, 2007*; *Doherty and McMahon, 2009*). Nevertheless, they are involved in a wide range of cellular processes, including cell migration (*Howes et al., 2010*), cell wounding responses (*Tam et al., 2010*), and pathogen internalization (*Vidricaire and Tremblay, 2007*), as well as membrane recycling in some neurons (*Gong et al., 2008*) and astrocytes (*Jiang and Chen, 2009*). Massive endocytic (MEND) responses described by us (*Fine et al., 2011*; *Hilgemann and Fine, 2011*; *Lariccia et al., 2011*) are the largest endocytic responses ever characterized. Large fractions of the plasmalemma that are internalized bind many amphipathic molecules less well than membrane remaining at the cell surface (*Hilgemann and Fine, 2011*). Thus, MEND internalizes preferentially the more 'ordered' portions of the surface membrane. Different forms of MEND can be distinguished on the basis of ATP-, polyamine-, and Ca-dependence (*Lariccia et al., 2011*). In addition, MEND can be initiated by rapidly cleaving sphingomyelin in the outer plasmalemma monolayer with bacterial sphingomyelinase C (*Lariccia et al., 2011*). It has been suggested that this form of MEND becomes activated during cell wound responses when native sphingomyelinases are translocated to the cell surface via exocytosis (*Tam et al., 2010*; *Corrotte et al., 2013*).

We describe here experiments that suggest an entirely different pathway by which MEND occurs in BHK fibroblasts subsequent to large Ca transients. These ATP-dependent endocytic responses are functionally similar to 'excessive' endocytosis that occurs after large Ca transients in secretory cells

**eLife digest** Cells use a process called endocytosis to absorb proteins and other molecules. There are many forms of endocytosis, but they usually involve the molecule of interest becoming tucked into a bud that forms in the cell membrane. This bud is then pinched off to leave the molecule inside a vesicle that is inside the cell. In general endocytosis is triggered by 'caging' proteins such as clathrin, but other forms are also possible. These "non-classical" forms of endocytosis are involved in processes as diverse as the internalization of pathogens and the response of cells to wounding, and sometimes they involve large fractions of the cell membrane being pinched off. Several different forms of "massive endocytosis" have been observed, but they have remained enigmatic in comparison to the classical forms of endocytosis.

Now Hilgemann et al. report a new pathway for massive endocytosis that is triggered by a sudden influx of calcium ions into Baby Hamster Kidney (BHK) fibroblasts. Up to 70% of the cell membrane can be pinched off during this process, especially areas of the membrane in which lipids and proteins are arranged in a more ordered pattern than in the average cell membrane. After massive endocytosis, it takes BHK cells about 30 minutes to replace the regions that were lost during the endocytosis.

Hilgemann et al. find that calcium ions exert their influence via mitochondria, which are the primary source of energy for most cells. In contrast to all other cell organelles, the mitochondria are surrounded by two concentric membranes. The influx of calcium ions causes pores in the inner membrane of the mitochondria, called permeability transition pores, to open so that coenzyme A, a small molecule that is required for fatty acid metabolism, is released into the cytoplasm of the cell. This is followed by the condensation of coenzyme A with a fatty acid and the attachment of fatty acids to the surface membrane proteins. The attachment of these fatty acids (a process known as palmitoylation) evidently promotes ordered regions of the cell surface to coalesce and be pinched off into the cytoplasm as membrane vesicles. A key unanswered question is whether the release of coenzyme A by mitochondria regulates biochemical processes in addition to endocytosis.

(*Smith and Neher, 1997*). Up to 70% of the cell surface of fibroblasts can be internalized, followed by replenishment of the plasmalemma from internal membrane pools over 20 to 40 min (*Lariccia et al., 2011*). To elucidate the cellular pathway by which MEND occurs, we attempted first to resolve how Ca promotes MEND, and second, to understand why more ordered membrane domains would be selectively internalized. As described here, our data support the hypothesis that Ca transients act initially to prime mitochondria to open PTPs (permeability transition pore) (*Giorgio et al., 2013*), releasing metabolites to the cytoplasm (*Azzolin et al., 2010*). At the distal end of the pathway, our data suggest that palmitoylation of surface membrane proteins promotes the coalescence of ordered membrane domains by promoting protein clustering, as described biochemically (*Levental et al., 2010*) and as occurs in anoxia-related metabolic stress (*Frank et al., 1980*).

That mitochondria might regulate surface membrane palmitoylation and thereby endocytosis is a novel hypothesis. Nevertheless, studies that suggest how this might occur have been available for decades. First, it is known that coenzyme A (CoA), which is synthesized on the outer mitochondrial surface, is accumulated into the matrix space by voltage-sensitive transporters (*Tahiliani, 1991*; *Leonardi et al., 2005*), generating CoA gradients to the cytoplasm of at least 50:1 (*Tahiliani and Neely, 1987*). Therefore, transient openings of nonselective mitochondrial pores, PTPs, can potentially release CoA and generate cytoplasmic CoA transients in the range of tens of micromolar without serious consequences for other mitochondrial metabolites. Second, the low micromolar concentration of free CoA in the cytoplasm has long been suggested to limit cytoplasmic acyl CoA synthetase activities (*Idell-Wenger et al., 1978*), whereas free CoA is less likely to limit pyruvate dehydrogenases that have higher CoA affinities (*Bremer, 1969*). Third, it is known that surface membrane proteins, visualized as membrane particles in freeze fracture studies, can cluster and decrease in number (*Frank et al., 1980*) in a cellular circumstance known to cause PTP openings, namely during anoxia and reoxygenation of cardiac tissue (*Brenner and Moulin, 2012*). Therewith, a working hypothesis is suggested that mitochondria might control palmitoylation of membrane proteins in response to metabolic stress by releasing CoA to the cytoplasm. In principle, this release may occur not only via PTP openings, but by

reverse CoA transport during mitochondrial depolarization caused by partial PTP openings, and via membrane defects that might occur during mitochondrial fission and fusion events in metabolic stress (*Knott et al., 2008*; *Ong et al., 2010*; *Frank et al., 2012*).

It is now well established that transient PTP openings occur physiologically, causing transient mitochondrial depolarizations accompanied by superoxide flashes (*Hausenloy et al., 2004*; *Saotome et al., 2009*; *Zhang et al., 2013*). One widely-held view is that low conductance PTP openings cause mitochondrial depolarization without loss of metabolites, and thereby release excess mitochondrial Ca to the cytoplasm (*Huser and Blatter, 1999*; *Korge et al., 2011*; *Zhou and O'Rourke, 2012*). Nevertheless, it appears possible for mitochondria to release metabolites by transient PTP openings without loss of mitochondrial membrane integrity or even global mitochondrial depolarization (*Dumas et al., 2009*). Knockout of cyclophilin D, a major PTP regulator (*Basso et al., 2005*), gives rise to intriguing phenotypes: hearts are mildly protected from ischemia/reperfusion damage, but animals are markedly less tolerant of physical exertion (swimming) and shift their energy metabolism from fatty acids to glucose (*Elrod et al., 2010*). From the prevailing interpretation of PTP function, these phenotypes will be the result of mitochondria Ca accumulation, driven by the failure of PTPs to open and release Ca when it accumulates in the matrix space. Our working hypothesis predicts that loss of mitochondrial depolarizations will also favor CoA accumulation by mitochondria and cause a decrease of cytoplasmic free CoA. Consequences of a lower cytoplasmic free CoA concentration might include the shift from fatty acid to glucose metabolism, owing to the different CoA affinities of acyl CoA synthetases and pyruvate dehydrogenases noted above.

With this background, we describe here multiple lines of evidence that mitochondrial responses to large Ca transients trigger MEND in BHK cells, that CoA and acyl CoAs are intermediates in the pathway, that membrane protein acylation is required, specifically via the acyl transferase, DHHC5, (*Li et al., 2010*), that PKCs and oxidative stress promote acyl CoA-dependent endocytosis without PTP openings, and finally that the MEND pathway may contribute significantly to constitutive surface membrane turnover in BHK cells.

## Results

### Evidence that MEND is initiated by mitochondria

Using BHK fibroblasts, data presented in *Figures 1 and 2* lends initial support to the hypothesis that palmitoylation can be initiated by release of CoA from mitochondria where it is accumulated by membrane potential-driven transporters (*Tahiliani, 1991*). Plasmalemma surface area is monitored as membrane electrical capacitance ($C_m$) via patch clamp of BHK cells that express cardiac Na/Ca exchangers, NCX1 (*Linck et al., 1998*). Potassium-free solutions are employed to block clathrin-dependent endocytosis (*Altankov and Grinnell, 1995*) and to minimize membrane conductance. Effects of square wave voltage perturbations (0.1–0.5 kHz) to determine $C_m$ are removed digitally from current records.

The 'Standard MEND' protocol is shown in *Figure 1A*. Patch clamp is established in a Ca-free extracellular solution using a cytoplasmic solution that contains a high concentration of Na (40 mM), ATP (8 mM) and GTP (0.2 mM). Then, cells are moved into one of multiple parallel flowing solution streams at 37°C, and Ca influx is activated by switching extracellular Ca from 0 to 2 mM Ca for 12 s. Membrane current (upper trace), which reflects 3Na/1Ca exchange (*Yaradanakul et al., 2008*), allows calculation of the absolute quantity of Ca entering cells. Since BHK cells are round after removal from dishes, we can also calculate cell volume from cell diameter (25–35 µm) and determine the absolute change of Ca occurring in cells. In the experiment of *Figure 1A*, the total cellular Ca concentration increases by 2.1 mM, and free Ca typically exceeds 50 µM in these experiments (*Lariccia et al., 2011*).

During Ca influx, membrane area (i.e., $C_m$, lower trace) increases by 35% as a result of the fusion of subplasmalemmal membrane vesicles to the plasmalemma (exocytosis). Then, after a delay of 30–120 s, membrane area begins to decline as a result of endocytosis until more than 50% of the plasmalemma is removed into vesicles just beneath the cell surface (*Lariccia et al., 2011*). As indicated below the experimental traces in *Figure 1A*, we quantified membrane area in relation to initial area at three points; just before Ca influx (1), after exocytosis (2), and after the MEND response (3). These values are subsequently presented as sets of three bar graphs. MEND responses, defined as the fractional decrease of membrane area between points 2 and 3, were compared only if Na/Ca exchange currents were not statistically different and exocytic responses in different groups were also similar.

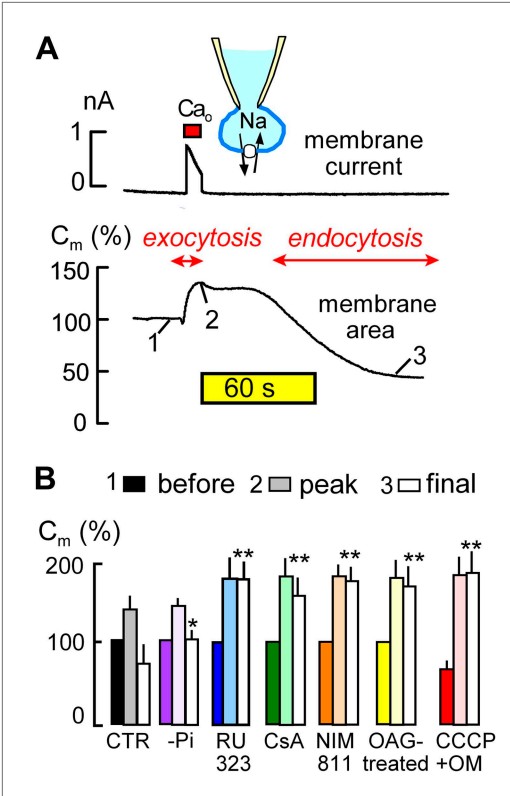

**Figure 1**. MEND is initiated by mitochondria. (**A**) Records of membrane current (above) and electrical membrane capacitance (below) during the Standard MEND protocol in BHK fibroblasts expressing cardiac Na/Ca exchangers (NCX1). Cells are opened in Ca-free extracellular solution with 0.5 mM EGTA, using a cytoplasmic (pipette) solution containing 40 mM cytoplasmic Na, 8 mM MgATP and 0.2 mM GTP. Thereafter, Ca influx via Na/Ca exchange is activated by extracellular application of 3 mM Ca for 10 s. During Ca influx, outward membrane current reflects 3Na/1 Ca exchange. Membrane area (i.e., $C_m$) increases by 30% as a result of exocytosis (i.e., fusion of vesicles to the cell surface). After terminating Ca influx, membrane area is stable for nearly 60 s and then declines over 2 min by 70% as plasmalemma is internalized via endocytosis. (**B**) Composite results implicating a role for mitochondria in the initiation of MEND. From left to right, bar graphs give results for Standard MEND (CTR, black and white), Standard MEND without cytoplasmic Pi (purple), with the mitochondrial Ca uptake inhibitor, RU323 (20 μM, blue), with the PTP blockers, cyclosporine A (5 μM, green) and NIM811 (2 μM, orange), after PKC activation by OAG (15 μM, yellow), and after rapid perfusion of a mitochondrial uncoupler, CCCP (20 μM) with an ATP synthetase inhibitor, oligomycin (OM, 5 μM, red). n > 6 in all panels. MEND was quantified as fractional decrease of $C_m$ from point 2 to point 3, and stars indicating significance have their usual meanings.

The first set of bar graphs in *Figure 1B* quantifies the Standard MEND response (Control, CTR; black and white). The second bar set (purple) shows that removal of inorganic phosphate (Pi, 1 mM) from the standard cytoplasmic solution reduces MEND responses to less than 30% of peak membrane area, as expected from the activating effect of Pi on PTPs (*Massari, 1996*). The third data set (blue) shows that the mitochondrial Ca uniporter inhibitor, RU323 (*Ying et al., 1991*) (15 μM), strongly blocks MEND. Subsequently, it is shown that the nonspecific PTP inhibitor, cyclosporine A (*Crompton et al., 1988*) (CsA, 5 μM in cytoplasmic and extracellular solutions; green) and the PTP/cyclophilin D-specific cyclosporine, NIM811 (N-methyl-4-isoleucine cyclosporine) (2 μM; orange) (*Waldmeier et al., 2002*), also strongly inhibit MEND. Importantly, the calcineurin inhibitor, FK506, had no effect on MEND, even at a high concentration (*Lariccia et al., 2011*). Activation of protein kinase C epsilon (PKCε) is reported to protect cells from oxidative damage by inhibiting PTP openings (*Baines et al., 2003*; *Budas and Mochly-Rosen, 2007*). Therefore, we incubated cells with a PKC activator, 1-oleoyl-2-acetyl-sn-glycerol (OAG, 15 μM), for 20 min before experiments, and Standard MEND was effectively blocked (yellow). Next, we performed experiments to induce rapid mitochondrial depolarization, which may be expected 'initially' to promote PTP openings (*Scorrano et al., 1997*), as well as to cause both reverse Ca transport (*Montero et al., 2001*) and reverse CoA transport (*Tahiliani, 1991*) from mitochondria. To do so, we included the mitochondrial uncoupler, CCCP (20 μM), and the ATP synthetase inhibitor, oligomycin (OM, 5 μM), in the pipette and opened cells only after placing them in a flowing solution at 37°C. As indicated by the solid red bar graph for data point '1' (before the Ca transient), cells underwent 30% MEND responses within 1 min after rupturing the membrane in the patch pipette (i.e., 'opening' the cell to the patch pipette). Thereafter, Ca influx initiated supernormal exocytic responses, from point '1' (before) to '2' (peak), but MEND was fully suppressed. Thus, while rapid mitochondrial depolarization with these agents can activate single MEND responses with very little delay, cells subsequently are refractory to MEND for the duration of patch clamp experiments (~30 min). We report lastly in this connection an experiment set in which we tested for inhibition of MEND by an antioxidant that becomes targeted to mitochondria, Mito-Tempo (*Jiang et al., 2009*). When included in the pipette solution at 10 μM, average MEND responses were reduced

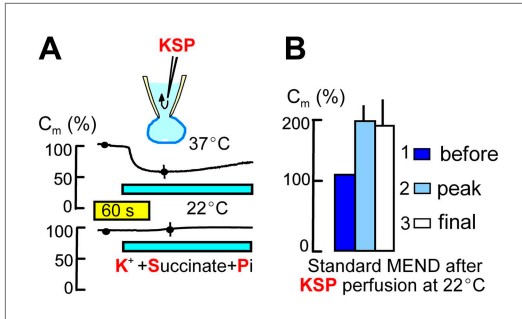

**Figure 2**. Metabolites that promote PTP openings rapidly initiate MEND. Experiments were started with Pi-free cytoplasmic solution. Subsequently, rapid internal perfusion of cells was initiated via a silica mico-capillary tube whose orifice was placed within 50 μm of the patch pipette orifice. KSP solution was prepared by replacing 80 mM NMG and 20 mM TEA in the standard internal solution with 100 mM K. In addition, 5 mM succinate and 1 mM Pi were added, and free Ca was buffered to 0.25 μM with 2 mM EGTA. (**A**) The upper record illustrates MEND responses recorded at 37°C upon cytoplasmic perfusion of KSP solution, and the lower record illustrates the failure of MEND to occur at 22°C. (**B**) Composite results for experiments in which the plasmalemma patch within the patch pipette was ruptured by suction (i.e., the cell was 'opened') at 22°C with KSP solution in the pipette, and the Standard MEND protocol was performed 1 to 2 min later at 37°C. n > 6 for all results.

from 42 ± 7% (n = 10) to 27 ± 6% (n = 8), but the difference was not significant.

*Figure 2* describes the initiation of MEND by perfusing the cytoplasm of cells rapidly with metabolic substrates that optimally open PTPs in isolated mitochondria at submicromolar free Ca (*Massari, 1996*). To do so, we place the end of a micro-capillary tube within 50 μm of the orifice of the patch pipette opening. Using positive pressure to generate rapid solution flow, diffusible molecules are exchanged between the cytoplasm and the pipette tip within 2–5 s (*Hilgemann and Lu, 1998*). As described in *Figure 2A*, we changed the standard potassium-free cytoplasmic solution as rapidly as possible to one with PTP-promoting constituents ('KSP' solution), namely succinate (5 mM), Pi (1 mM), free Ca set to 0.25 μM with 2 mM EGTA, and potassium (100 mM). After a delay of several seconds, MEND responses occurred over a time course of somewhat less that 1 min and amounted to 34 ± 6% of initial cell area at 37°C. As shown by the lower trace in *Figure 2A*, the same protocol induced no response at 22°C. Nevertheless, *Figure 2B* shows that Standard MEND responses at 37°C were negligible in cells that were initially opened at 22°C with KSP solution (potassium/succinate/phosphate-containing solution) in the pipette. Thus, KSP solution, like mitochondrial depolarization, induces a MEND-refractory state within at most 1 min, even when no MEND occurs at 22°C. Given that cytoplasmic solutes exchange with pipette solutions within a few seconds, this result suggests that metabolites required for MEND might be released and lost into the patch pipette during cytoplasmic application of KSP solution.

## MEND requires acyl CoA synthesis and palmitoylation

Initial evidence that MEND requires generation of acyl CoAs and palmitoylation is presented in *Figure 3*. The first data set (black and white) shows that Standard MEND is normal in cells that were incubated for 1 hr with 1:1 palmitate-loaded albumin (50 μM; 'Alb+FA'). The second data set (purple) shows that MEND is suppressed in cells that were incubated with fatty acid (FA)-free albumin ('Alb-FA', 50 μM) for 1 hr before experiments to deplete cellular fatty acids. The next data sets show that MEND is inhibited 70% by the palmitoylation inhibitor, 2-bromopalmitate (BP, 50 μM; blue) (*Jennings et al., 2009*), is decreased 50% by the acyl CoA synthetase inhibitor, triacsin C (TrC, 2 μM; green) (*Omura et al., 1986*) when it is applied acutely in both the pipette and extracellular solutions, and MEND is fully blocked by acute application of both bromopalmitate and triascin C (orange). Thus, both synthesis of acyl CoAs and palmitoylation may be required in the reaction pathway leading to MEND.

In subsequent experiments, we examined two opposed ways in which CoA can be predicted to modulate the MEND pathway, assuming that CoA is first used to generate acyl CoA from fatty acid and subsequently is released during palmitoylation of membrane proteins. First, we tested whether free CoA is in fact an intermediate in the pathway. To do so, we perfused cells with a high concentration of a CoA-metabolizing enzyme, namely human acetyl CoA synthetase (ACS, 20 μM; $K_d$ for CoA, 11 μM [*Luong et al., 2000*]) with acetate (Ace, 6 mM) to rapidly convert CoA to acetyl CoA. Acetate itself did not affect MEND (pink), but acetate together with acetyl CoA synthetase blocked MEND (yellow). Second, we tested whether a very high concentration of CoA (3 mM) would inhibit MEND, as expected for product inhibition of acyl transferase reaction that releases free CoA. As shown in the last data set, CoA at a concentration of 3 mM strongly inhibited MEND (red), and *Figure 3B* documents that acyl

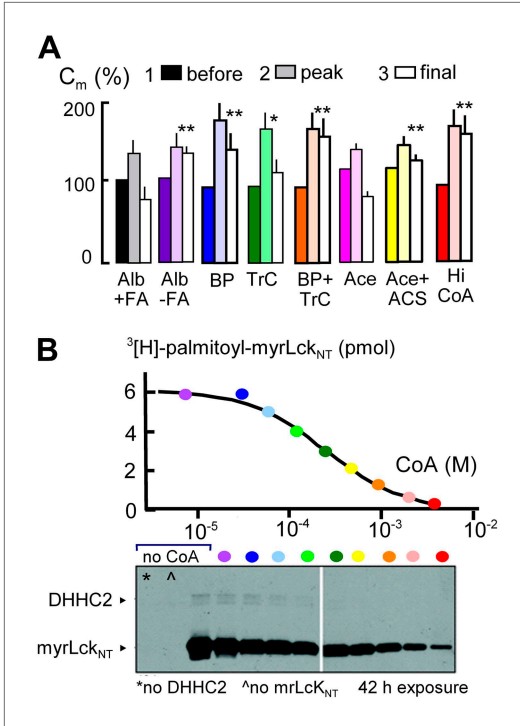

**Figure 3**. MEND is blocked by preventing protein palmitoylation reactions. (**A**) From left to right, composite results for Standard MEND after incubating cells with 1:1 palmitate-loaded albumin (50 µM) for 1 hr (black), after incubating cells with fatty acid-free albumin (Alb, 50 µM) for 1 hr (purple), with the palmitoylation inhibitor, bromopalmitate (BP, 50 µM) included in all solutions (blue), with the acyl CoA synthetase inhibitor, Triascin C (TrC, 2 µM) in all solutions (green), with BP (50 µM) and TrC (2 µM) in all solutions (orange), with cytoplasmic acetate (Ace, 6 mM; pink), with acetate (6 mM) and acetyl CoA synthetase (ACS, 20 µM; yellow), and with a high cytoplasmic CoA concentration (3 mM) to block DHHCs (red). For all results, n > 6. (**B**) CoA inhibition of DHHC2-mediated palmitoylation of the N-terminus of myristoylated lymphocyte-specific kinase (myrLckNT). For details, see 'Materials and methods'.

transferase activity of the DHHC2 transferase is effectively blocked by 3 mM CoA. Inhibition occurs with a $K_i$ of about 0.2 mM, and activity is reduced by more than 90% with 3 mM CoA.

## Cytoplasmic perfusion of acyl CoA or CoA circumvents MEND block by four mitochondrial mechanisms

*Figure 4* demonstrates that MEND blockade via the mitochondrial mechanisms just described is entirely overcome by perfusion of myristoyl CoA (mCoA) or CoA into cells. We employed mCoA instead of palmitoyl CoA because its lower affinity for membranes is advantageous to achieve rapid cytoplasmic concentration changes via the pipette perfusion technique. In *Figure 4A*, cyclosporine A (3 µM) was used to block MEND. Ca influx caused a 35% exocytic response, as usual, and membrane area ($C_m$) was then stable until mCoA (15 µM) was perfused into the cell. Within seconds, membrane area began to fall, and within 2 min nearly 70% of the cell surface was removed by endocytosis. *Figure 4B* shows composite data for mCoA release of MEND block by cyclosporine A (black and white), followed by results for release of MEND block by CCCP and oligomycin (purple), KSP-induced MEND block (blue), and OAG-induced MEND block (green). The last two data sets in *Figure 4B* show that perfusion of CoA (20 µM), instead of mCoA, overcomes cyclosporine A MEND block equally well as mCoA (yellow), and finally that MEND block by a high concentration of CoA (3 mM) is immediately relieved by its washout from cells (red), as expected for a simple product inhibition mechanism.

## The acyl tranferase, DHHC5, is required for MEND in BHK cells

We next address the role of DHHC acyl transferases that can mediate surface membrane protein palmitoylation in MEND. From more than 20 DHHC transferases (*Mitchell et al., 2006*) only DHHC2 and DHHC5 are known at this time to traffic to and become active at the plasmalemma (*Greaves et al., 2011*; *Li et al., 2011*; *Thomas et al., 2012*), and DHHC5 may be more ubiquitous (*Li et al., 2011*). Using siRNA for DHHC5 with lipofectamine2000 transfection, *Figure 5A* documents in Western blots that DHHC5 expression was decreased by >80% in BHK cells. However, the magnitudes of Standard MEND responses became variable after transfection protocols using either lipofectamine2000 or RNAmax. We were able to overcome this variability and generate even larger MEND responses by incubating BHK cells during the entire siRNA protocol with a low concentration of a PKC inhibitor, staurosporine (STS, 0.1 µM), that prevents PKC-induced inhibition of PTPs (*Ytrehus et al., 1994*). Cells were returned to STS-free solutions 5 min before experiments. As shown in *Figure 5B* using LF2000 to transfect cells, exocytic responses in control transfected cells were larger than normal, and subsequent MEND responses removed fully 75% of the cell surface. In cells transfected with siRNA for DHHC5 (right data set in *Figure 5B*), these MEND responses were decreased on average to 25% of the cell surface, corresponding to a 66% decrease of the MEND response. As shown in *Figure 5C*, siRNA of DHHC5 also strongly

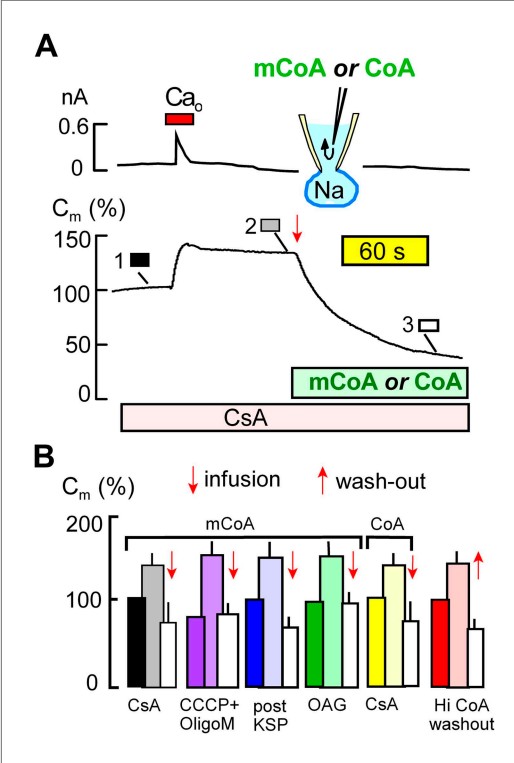

**Figure 4**. Cytoplasmic perfusion of acyl CoA or CoA circumvents four mitochondrial MEND blocks. (**A**) Typical experiment in the presence of cyclosporine A (CsA, 3 μM) to block MEND. Ca influx cause a 40% increase of $C_m$ via exocytosis, and $C_m$ remains stable thereafter for minutes. Cytoplasmic perfusion of myristoyl CoA (mCoA, 15 μM) via the micro-capillary within the patch pipette causes a MEND response that begins within 10 s and internalizes 70% of the plasmalemma within 2.5 min. (**B**) Composite results quantifying MEND that occurs when mCoA or CoA is perfused into the cytoplasm of cells in which MEND has been blocked by interventions acting on mitochondria. From left to right, bar graphs present results for pipette perfusion of mCoA (15 μM) in cells in which MEND was blocked by cyclosporine A (CsA, 3 μM; black and white), by opening cells at 22°C with CCCP (20 μM) and oligomycin (5 μM; purple), by opening cells at 22°C with KSP (blue), and by pretreatment of cells with OAG (15 μM) for 30 min (green). The penultimate results quantify MEND caused by pipette perfusion of CoA (20 μM) into cyclosporine A-blocked cells (yellow). The final data set shows results for cells in which MEND was blocked by a high cytoplasmic CoA concentration (3 mM). Ca influx caused on average 38% exocytic responses, and $C_m$ was then stable. When CoA was perfused out to the cytoplasm ('wash-out'), endocytosis started within 15 s and amounted to 56% of the plasmalemma on average. For all results, n > 6.

blocked MEND responses induced by pipette perfusion of mCoA into cells that were MEND-blocked by cyclosporine A (3 μM). *Figure 5D* quantifies KSP-induced MEND in BHK cells that were not STS-treated. Accordingly, KSP-induced MEND in cells transfected with scrambled DHHC5 siRNA amounts to only 14%. Nevertheless, the responses were reproducible, and they were highly significantly reduced to 2.5% by DHHC5 siRNA transfection.

## Final steps of MEND: activation by PKCs, ROS transients, and cargo

As established in *Figures 3 and 4*, the generation of acyl CoA is required for the occurrence of MEND in these experiments. As shown in *Figure 6A*, however, the introduction of acyl CoA (mCoA) into cells is not sufficient to initiate MEND. Under standard conditions, cytoplasmic perfusion of mCoA (15 μM) causes only 5 to, at most, 10% endocytic responses over 10 min (n > 20; green square in *Figure 6A*). Therefore, Ca must be acting by at least one additional mechanism for MEND to occur. From many possibilities, we describe in *Figure 6* that two common Ca-dependent cell signals, activation of PKCs and generation of reactive oxygen (ROS), both can promote MEND in the presence of acyl CoA and in the absence of Ca transients.

*Figure 6A* presents results for applying the same PKC activator, OAG, used to inactivate MEND 'prior' to experiments in *Figure 1*. As shown by the dashed line, application of OAG (15 μM) on its own causes very little or no endocytosis. However, when OAG (15 μM) is applied for 1.5 min in the presence of mCoA (15 μM) in the pipette, membrane area begins to decrease with a delay of about 30 s and continues to decrease for several minutes after removal of OAG. As shown in *Figure 6B*, application of $H_2O_2$ (hydrogen peroxide) (80 μM) for 4 min has little or no effect in the absence of mCoA in the cytoplasm. In the presence of mCoA (15 μM), application of $H_2O_2$ causes little or no change initially, but upon its removal membrane area begins to decline robustly within 1 min and on average more than 25% of the cell surface is lost over 10 min. This pattern suggests that ROS have an acute inhibitory effect on MEND, which is rapidly relieved when $H_2O_2$ is removed. The stimulatory effect of ROS on MEND, however, is a long-lived effect. The results suggest that transient generation of ROS during the MEND protocol promotes the final steps of MEND, subsequent to generation of acyl CoA.

In *Figure 6C* we provide further details about the PKC and $H_2O_2$ effects, quantifying the occurrence of MEND as the average rate of membrane loss over 4 min. MEND responses promoted by OAG are

**eLIFE** Research article

Biophysics and structural biology | Cell biology

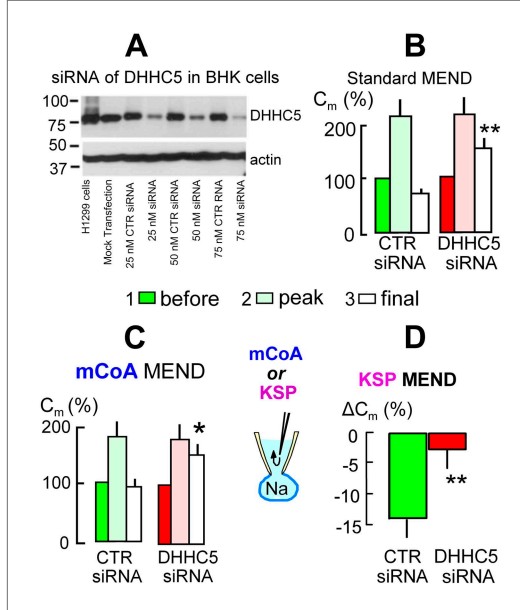

**Figure 5**. The acyl tranferase, DHHC5, is required for MEND in BHK cells. (**A**) DHHC5 knockdown by siRNA to ZDHHC5. BHK cells were transfected with control siRNA or ZDHHC5-specific siRNA at indicated concentrations using Lipofectamine 2000. 72 hrs after transfection, cells were harvested and cell lysates were processed for Western blot analysis using 20 μg protein per lane. A human non-small lung carcinoma cell line, H1299, was used as a positive control for anti-ZDHHC5 antibody. (**B**) MEND in BHK cells amounts to 75% of the cell surface after growing cells with staurosporin (0.1 μM), and DHHC5 siRNA decreases MEND by 63% (p<0.01). (**C**) mCoA-relief of MEND block by cyclosporine A (3 μM) is reduced by 75% (p<0.05) in BHK cells transfected with siRNA for DHHC5 vs scrambled siRNA. (**D**) KSP-induced MEND in BHK cells is reduced from 14 to 2.5% by DHHC5 siRNA transfection using RNAmax (p<0.01). For all results, n > 6.

strongly inhibited by a peptide substrate inhibitor of conventional PKCs (PKC peptide (19–36), 1 μM), confirming that conventional PKCs are involved (*House and Kemp, 1987*). By contrast, the induction of MEND by $H_2O_2$ is not inhibited by the PKC peptide, indicating that $H_2O_2$ is not acting through the PKCs that can promote MEND. Finally, we show that MEND activation by $H_2O_2$ and OAG are not additive; the activation of MEND by one reagent precludes a further effect by the other.

It is beyond the scope of this article to resolve how PKCs and $H_2O_2$ promote MEND. However, one possibility raised by the literature is that phosphorylation of some DHHC protein substrates by PKCs can be permissive for their subsequent palmitoylation. This is the case for the dually palmitoylated, Na/K pump regulatory protein, phospholemman (PLM) (*Tulloch et al., 2011*), which is expressed preferentially in muscle (*Geering, 2006*). With this background, we next generated a T-REx-293 cell line in which expression of PLM can be activated in a tetracycline-dependent manner. As shown in *Figure 6D*, we quantified the rate at which MEND occurred over 4 min after a 1 min activation of PKCs by OAG (15 μM) in the presence of cytoplasmic mCoA (15 μM). The overexpression of PLM caused a three-fold increase in the rate of PKC/mCoA-dependent MEND. This result clearly supports the idea that MEND can be promoted by the presence of proteins that can be palmitoylated and therefore is a 'cargo-dependent' form of endocytosis. At this time, we can only speculate that long-term protein modifications caused by transient oxidative stress also promote the availability of membrane protein palmitoylation sites.

## The MEND pathway may be constitutively active

Given that MEND can be activated without large Ca transients, the question is raised whether the MEND pathway may be constitutively active at the cytoplasmic acyl CoA concentrations occurring normally in cells. We describe in *Figure 7* experiments that lend initial support for this possibility by analyzing surface membrane changes in response to treating BHK cells with the putatively specific cyclosporine D/PTP inhibitor, NIM811 (2 μM). As shown in *Figure 7A*, BHK cells are essentially round after removal from dishes. From micrographs, we measured the diameter of each cell in both the 'X' and 'Y' directions, calculated the spherical area of each cell, and determined the $C_m$ of each cell after opening it via patch clamp. In the first group of experiments, BHK cells were removed from dishes and were incubated for 1 or 3 hr, either with or without NIM811 at 37°C. Results for control cells were pooled, giving three groups of measurements. The average diameters of cells employed (28.2 ± 0.9, 27.8 ± 1.0 μm, and 29.9 ± 1.4 μm) were very similar. Assuming a specific membrane capacitance of 1 pF per 100 μm², we calculated for each cell the ratio of surface membrane area to the area of a sphere with the cell diameter. For control cells, a ratio of 2.2 ± 0.1 was obtained, indicating that surface membrane undulations account for one-half of the cell surface. After 1 hr of NIM811 treatment this ratio was not changed, but the ratio increased significantly to 2.6 ± 0.1 after 3 hr of incubation. In the

Hilgemann *et al*. eLife 2013;2:e01293. DOI: 10.7554/eLife.01293

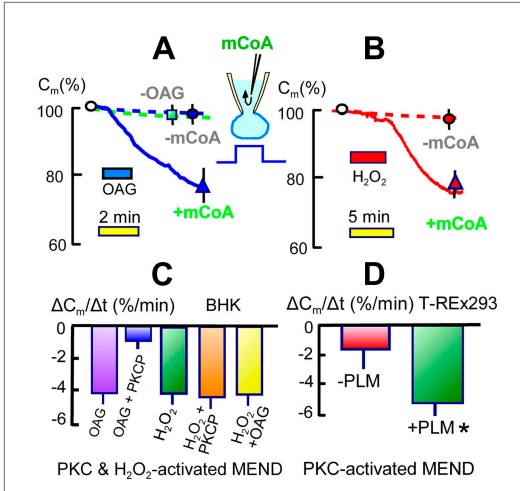

**Figure 6**. Activation of acyl CoA-dependent MEND without Ca transients. (**A**) Pipette perfusion of mCoA (15 µM) without previously activating a Ca transient causes little or no endocytosis over 5 min (green square). Extracellular application of OAG (15 µM) for 1 to 2 min also has little or no effect, even after 6 min. With mCoA (15 µM) in the cytoplsamic solution, OAG initiates a decrease of membrane area that continues for several minutes after OAG is removed, amounting to 24% on average after 6 min. (**B**) Application of $H_2O_2$ (80 µM) for 4 min has little or no effect over 15 min in the absence of mCoA. With mCoA (15 µM) in the cytoplasmic solution, however, a large decrease of membrane area occurs when $H_2O_2$ is removed, amounting to 26% on average after 6 min. (**C**) MEND responses quantified as percent decrease of membrane area per min over 4 min. From left to right, the bar graphs quantify MEND caused by OAG, inhibition of OAG-induced MEND by PKC peptide 19–36 (1 µM in the pipette), MEND caused by $H_2O_2$, lack of effect of PKC peptide (19–35) on $H_2O_2$ MEND, and average MEND responses to $H_2O_2$ and OAG applied sequentially. (**D**) MEND in T-Rex-293 cells with inducible PLM expression. The bar graphs quantify the percent decrease of membrane area over 4 min after applying OAG. OAG-activated MEND is increased nearly three-fold when PLM expression has been induced for 24 hr.

second set of experiments, we grew BHK cells normally for 28 hr in the presence of NIM811 (2 µM) and carried out equivalent measurements after removing cells from dishes. Membrane area in cells of equal size was increased on average by 27% (p<0.01). In summary, treatment of BHK cells with NIM811 causes a significant increase of membrane area after a delay of 1 hr, reaching a plateau within about 3 hr, indicating 'either' that membrane insertion at the cell surface is increased 'or' that membrane removal from the cell surface is decreased, as expected for partial inhibition of endocytosis.

## Discussion

Patch clamp techniques allow extensive manipulation of single cells, and we have exploited those possibilities to analyze Ca-dependent endocytosis that occurs in BHK fibroblasts by unconventional mechanisms (*Lariccia et al., 2011*). While our methods allow manipulation of many variables that cannot be controlled in classical biochemical experiments, they do not allow measurements of metabolites and/or other intermediates in parallel. Therefore, this study relies on manipulating the complete cell environment. In an accompanying article (*Lin et al., 2013*), we address the MEND pathway in cardiac muscle using additionally biochemical and optical methods.

In spite of clear limitations, we argue that our data sets support working hypotheses summarized in *Figure 8*: transient openings of mitochondrial PTPs, which can be inhibited by the activation of PKCε (*Baines et al., 2003*), rapidly release CoA from the mitochondrial matrix space to the cytoplasm. Then, generation of cytoplasmic acyl CoA promotes membrane protein palmitoylation via the DHHC5 acyl transferase. We hypothesize that palmitolyation promotes membrane protein clustering (*Levental et al., 2010*) and that clustering beyond a threshold promotes budding and internalization of *Lo* membrane domains (*Fine et al., 2011*; *Hilgemann and Fine, 2011*; *Lariccia et al., 2011*). We speculate that conventional PKC activation and 'transient' $H_2O_2$ stress promote MEND progression by increasing the availability of sites for palmitoylation, whereas the immediate presence of oxidative stress inhibits MEND progression by inhibiting palmitoylation reactions. Our data does not address how the final steps of endocytosis take place. We have shown previously that $PIP_2$ must be resynthesized after its depletion during Ca transients (*Lariccia et al., 2011*), but we do not know what specific role phosphoinositide turnover plays. Other issues to be clarified include the roles of peripheral membrane proteins that associate with *Lo* membrane, such as flotillins (*Otto and Nichols, 2011*) and annexin-2 (*Chasserot-Golaz et al., 2005*), as well as the potential of clustered proteins with large cytoplasmic domains, for example, Na/K pumps, to bend membranes and promote budding. We summarize next the experimental support for these hypotheses in relation to specific questions that the hypotheses raise.

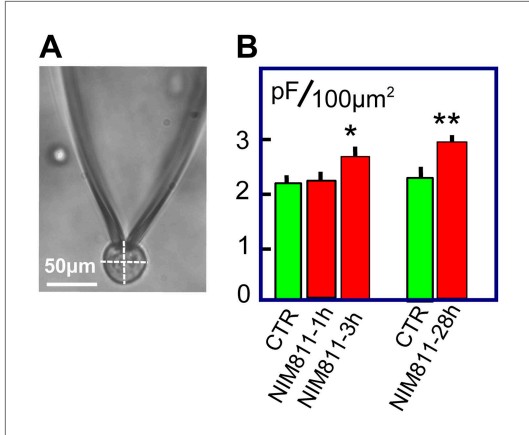

**Figure 7**. NIM811 increases surface area of cultured BHK cells. (**A**) Micrograph of BHK cell during patch clamp. 25× LWD lens. Cell diameters were calculated as the average of a horizontal and vertical line transecting each cell, as indicated. The spherical area of each cell was calculated, and the ratio of the electrically determined area to the spherical area was calculated assuming 1pF to be 100 $\mu m^2$. (**B**) First three bar graphs give results for BHK cells removed from dishes and incubated at 37°C for 1 or 3 hr without (CTR) and with NIM811 (2 $\mu M$). Results for control cells were not significantly different and were pooled. The fourth and fifth bar graphs give results for control cells (CTR) and cells grown with NIM811 (2 $\mu M$) for 30 hr.

## Do mitochondrial PTPs open before MEND occurs?

That mitochondria are involved in the initiation of MEND is supported by results for six interventions expected to act via mitochondria (*Figures 1 and 2*). The following results support the idea that PTP openings are a prerequisite for Ca-activated MEND during the standard protocol: (1) reproducible MEND, constituting 50% of the plasmalemma, requires cytoplasmic Pi, an activator of PTPs (*Massari, 1996*). (2) MEND is blocked by inhibiting Ca uptake by mitochondria, a mechanism that will stop both mitochondrial depolarization and PTP openings. (3) MEND is blocked by two cyclosporines, one considered a specific cyclophilin D/PTP inhibitor (*Waldmeier et al., 2002*). Although cyclosporine A also inhibits calcineurin, we determined previously that calcineurin inhibition by FK506 does not inhibit MEND (*Lariccia et al., 2011*). (4) MEND can be blocked by prior PKC activation, consistent with suggestions that PKCs protect cells from reperfusion injury by inhibiting PTP openings (*Ytrehus et al., 1994*). (5) MEND occurs when cells are internally perfused with mitochondrial substrates that support PTP openings, namely succinate with Pi in potassium-rich cytoplasmic solution at an optimal free Ca concentration (0.2 $\mu M$, [*Massari, 1996*]). These responses (*Figure 2*) occur without an initial exocytic phase, indicating that Ca release from mitochondria, if it occurs, does not cause large Ca transients in these experiments. (6) Finally, we bring to bear original descriptions of PTP openings, monitored via mitochondrial light scattering (*Haworth and Hunter, 1979*). Under the conditions of those experiments, the rate at which PTPs open increases over the Ca concentration range of 10–500 $\mu M$, but maximal responses are still obtained with 30 $\mu M$ free Ca in less than 10 s. Free cytoplasmic Ca concentrations in our experiments definitively exceed 50 $\mu M$ (*Lariccia et al., 2011*), and we activate Ca influx for 10 to 14 s in the Standard MEND protocol. Therefore, it is very likely that full PTP pore openings occur during these experiments.

## Must PTPs open fully or does mitochondrial depolarization suffice to initiate MEND?

Rapid cytoplasmic application of KSP solution or CCCP with oligomycin in BHK cells induces a MEND-refractory state (*Figure 2* and *Figure 1B*), which we interpret to be the consequence of CoA release via PTP openings with subsequent loss of CoA into the pipette. We admit in this regard that CCCP and oligomycin have not previously been shown to cause transient PTP openings when rapidly introduced into cells. However, depolarization definitively promotes PTP openings (*Scorrano et al., 1997*). In fact, more specific explanations may emerge because recent work indicates that PTPS are formed by dimers of the target of oligomycin, the ATP synthetase (*Giorgio et al., 2013*).

Related to these experiments, it is an important question whether mitochondrial depolarization per se causes substantial CoA release via reverse CoA transport. Studies of cardiac mitochondria indicate that reverse CoA transport can decrease matrix CoA with a half-life on the order of 15 min when mitochondria are depolarized with valinomycin (*Tahiliani, 1991*). Accordingly, reverse transport of CoA by depolarized mitochondria would release only about 10% of the mitochondrial CoA content within the delay of 1–2 min that precedes the onset of MEND after a Ca transient. This may well be significant in intact cells. However, the fact that KSP solution effectively causes a MEND-refractory state within 1 min at room temperature (*Figure 2*) is much more consistent with rapid CoA loss via diffusion through a pore than via the reverse transport process.

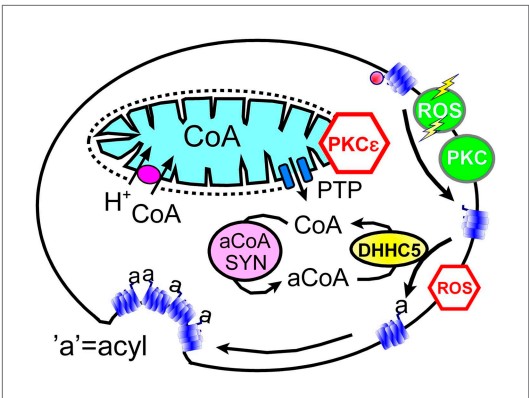

**Figure 8**. Hypothetical MEND pathway. Mitochondria accumulate CoA via voltage-dependent transporters that are probably proton-coupled (*Tahiliani, 1989*). CoA can be released in response to PTP openings, either directly or more slowly via reverse CoA transport during mitochondrial depolarization. During a MEND response the activation of PKCs will restabilize mitochondria and promote MEND progression at the cell surface. PTP openings are inhibited when PKCs are activated 'prior' to the MEND protocol. Acyl CoA transients occur upon release of CoA because acyl CoA synthetases are CoA-limited (*Idell-Wenger et al., 1978*). We speculate that conventional PKCs and 'transient' oxidative stress increase the availability of palmitoylation sites at the surface membrane, whereas the immediate presence of ROS inhibits palmitoylation reactions.

## Is MEND preceded by a burst of acyl CoA synthesis and membrane protein palmitoylation?

That the pathway to Ca-activated MEND in BHK cells requires generation of acyl CoAs and palmitoylation of proteins at the cell surface is supported by the following evidence: (1) fatty acid depletion of cells for 1 hr, a specific inhibitor of acyl CoA synthetase activity (*Omura et al., 1986*), and a nonspecific inhibitor of palmitoylation, bromopalmitate, all reduce MEND, and triascin C with bromopalmitate causes complete MEND block. (2) An acetyl CoA synthetase, when perfused into cells together with acetate to deplete CoA, also fully blocks Ca-activated MEND. (3) A high concentration of CoA, which fully blocks acylation activity of a DHHC acyl transferase, fully blocks Ca-activated MEND. (4) When MEND is blocked by four different interventions that probably act through mitochondria, introduction of acyl CoA into the cytoplasm fully restores MEND. (5) Low concentrations of CoA are equally effective as acyl CoA in promoting MEND in these protocols. (6) High CoA concentrations are demonstrated to block DHHC acyl transferase activity and to strongly block MEND, and this block is rapidly reversed by perfusion CoA out of the cytoplasm. (7) Ca-activated MEND, MEND induced by acyl CoA in cyclosporine A-blocked cells, and MEND induced by PTP-promoting substrates (KSP solution) are all selectively inhibited by knockdown of the DHHC5 acyl transferase, which is known to be active at the surface membrane (*Li et al., 2011*). As with evidence for mitochondrial involvement, multiple explanations are possible for individual outcomes. However, the results together provide clear evidence that Ca-activated MEND requires acyl CoA synthesis and acyl transferase activity, that is protein palmitoylations, within the time frame in which MEND occurs. While palmitoylation within the secretory pathway clearly can target proteins to the plasmalemma (*Greaves et al., 2009*), the present data sets clearly suggest that progressively greater palmitoylation of surface membrane proteins can promote removal of proteins from the cell surface by endocytosis. Whether or not MEND requires palmitoylation of specific proteins is not addressed by the present data sets. However, we have demonstrated in *Figure 6D* that the presence of a membrane protein that can be dually palmitoylated in a PKC-dependent manner (*Tulloch et al., 2011*) facilitates the occurrence of MEND. This clearly suggests that MEND is a 'cargo-dependent' endocytic process.

## Do sphingomyelinases play a role in Ca-activated MEND?

It seems likely that Ca-activated MEND in BHK cells is mechanistically related to endocytosis that occurs during cell wounding (*Tam et al., 2010*). As noted in the Introduction, it is proposed that exocytosis during cell wounding translocates sphingomyelinases to the cell surface, where they subsequently generate ceramide and promote endocytosis (*Tam et al., 2010*). Further, it is proposed that caveolae are internalized in a specific manner (*Corrotte et al., 2013*). From our perspective, it is certain that several different mechanisms can cause MEND (*Lariccia et al., 2011*), and extracellular application of sphingomyelinase C does so very effectively (*Lariccia et al., 2011*) when about 50% of cellular sphingomyelin is cleaved (*Van Tiel et al., 2000*). Our tests for involvement of native sphingomyelinases in BHK MEND responses were entirely negative, and we were able to dissociate exocytosis from MEND (*Lariccia et al., 2011*). While caveolae might well become involved in MEND, the loss of 70% of the cell surface during large MEND is inconsistent with the much smaller membrane area

represented by caveolae in almost all cells (*Bretscher and Whytock, 1977*; *Parton and del Pozo, 2013*). In our view, therefore, MEND involves the internalization of ordered membrane domains that can form via phase separation from bulk membrane as the result of multiple synergistic cellular processes, one of them being progressively more extensive palmitoylation of membrane proteins that appears to occur in Ca-activated MEND in BHK cells.

### Is the MEND pathway constitutively active?

Constitutive endocytosis in BHK cells has been analyzed rather extensively (*Marsh and Helenius, 1980*; *Griffiths et al., 1989*). Between one and two percent of the surface membrane is internalized per min, mostly via well-defined coated pits (*Marsh and Helenius, 1980*). Nevertheless, the precise quantitation of different endocytic forms does not seem secure. Certainly, endocytic mechanisms that internalize 'lipid raft' proteins, in particular GPI-anchored proteins (*Lakhan et al., 2009*), are also well characterized in BHK cells (*Fivaz et al., 2002*; *Refaei et al., 2011*). It therefore cannot be discounted that the mechanisms underlying MEND are constitutively active, possibly in dependence on spontaneous mitochondrial depolarizations and superoxide flashes (*Elrod et al., 2010*; *Zhang et al., 2013*) and consistent with the increase of sarcolemma area caused by NIM811 (*Figure 7*). MEND can internalize surface membrane at a rate of 50% per min (*Figure 1*), at least 25-times greater than constitutive membrane turnover. Therefore, a constitutive activity of the MEND pathway equal to even one percent of its maximal activity would be a substantial endocytic flux in BHK cells.

Independent of mitochondrial involvement, our experiments raise a question whether the distal part of the MEND pathway, that is acyl CoA/palmitoylation-dependent endocytosis, might be a form of 'lipid raft'-dependent endocytosis that is physiologically activated by PKCs and/or oxidative stress. Clearly, DHHC5 is constitutively functional in cells (*Li et al., 2011*; *Thomas et al., 2012*), and both PKC activation and $H_2O_2$ promote acyl CoA-dependent endocytosis without the activation of Ca transients (*Figure 6*). Endocytosis following PKC activation might in principle reflect the endocyotic process that normally inactivates PKCs by removing PKCs from the cell surface (*Carmena and Sardini, 2007*). Although generally ascribed to conventional, ubiquitin-dependent endocytosis, a 'caveolar' or 'lipid raft' pathway that is blocked by nystatin has also been shown to participate in PKC inactivation (*Leontieva and Black, 2004*).

Our work suggests that the dually palmitoylated membrane protein, PLM, can be used as a simple model to study PKC-dependent endocytosis in future work, independent of mitochondrial involvement. PLM appears to be palmitoylated in dependence on its phosphorylation by PKCs (*Tulloch et al., 2011*), and expression of PLM promotes acyl CoA/PKC-dependent endocytosis (*Figure 6*). Therefore, mutations of PLM to modulate selectively its palmitoylation and phosphorylation status should allow insightful experiments. It appears more challenging to elucidate how transient $H_2O_2$ oxidative stress activates endocytosis, although an obvious starting point will be to determine how the palmitoylation status of membrane proteins is changed by transient oxidative stress.

In summary, our work lends support to a working hypothesis that mitochondrial signaling impacts the plasmalemma by controlling and/or modulating surface membrane protein palmitoylation and subsequently palmitoylation-dependent endocytosis. One trigger appears to be release of CoA from mitochondria during mitochondrial depolarizations that involve transient PTP openings (*Huser and Blatter, 1999*; *Korge et al., 2011*). Activation of PKCs prior to a Ca transient blocks MEND, while both PKCs and transient generation of ROS support MEND progression at the cell surface. Accordingly, a wave of PKC activation during MEND will tend to restabilize mitochondria but promote ongoing endocytic events. How MEND is related to different categories of 'lipid raft'-dependent endocytosis delineated to date (*Lajoie and Nabi, 2007*; *Mayor and Pagano, 2007*) remains to be established. It will be of great interest to establish how and if the MEND pathway is related to Ca-activated excessive endocytosis (*Smith and Neher, 1997*) and bulk endocytosis (*Cousin, 2009*) in secretory cells.

## Materials and methods

### Electrical methods, cell cultures, and myocytes

Patch clamp (*Yaradanakul et al., 2008*) and cell cultures (*Linck et al., 1998*) were as described (*Lariccia et al., 2011*). Effects of voltage pulses (0.1–0.5 kHz) to determine $C_m$ are filtered out of records. We employed relatively large BHK cells (40–120 pF) because MEND was more reliable in that subpopulation. A T-REx-293 cell line with inducible PLM expression was prepared as follows: PLM was excised from

pAdTrack-CMV-PLM (a gift from Lois Carl, Pennsylvania State University) and then inserted into pcDNA5/FRT/TO vector (Life technologies) using EcoRV and HindIII restriction sites. Flp-In T-REx 293 cells were transfected with pcDNA5/FRT/TO-PLM and pOG44 and then selected with hygromycin to generate cells expressing PLM in the tetracycline inducible system.

## Solutions

Standard MEND solutions minimize all currents other than NCX1 current. Extracellular solution contained in mM: 120 n-methyl-d-glucamine (NMG), 4 $MgCl_2 \pm 2$ $CaCl_2$, 0.5 EGTA, 20 TEA-OH, 10 HEPES, pH 7.0 with aspartate. Cytoplasmic solution contained in mM: 75 NMG, 20 TEA-OH, 15 HEPES, 40 NaOH, 0.5 $MgCl_2$, 0.8 EGTA, 0.25 $CaCl_2$, 1 Pi set to pH 7.0 with aspartate. Unless stated otherwise, 8 mM MgATP, 2 mM TrisATP, and 0.2 mM GTP were employed in cytoplasmic solutions with a free Mg of 0.5 mM. KSP cytoplasmic solution contained 110 KOH, 40 NaOH, 10 histidine, 2.0 EGTA, 0.45 $CaCl_2$, nucleotides as just given, and pH 7.0 with aspartate. Bath solution in NIC recording and FITC-dextan uptake experiments contained in mM: 120 NaCl, 5 KCl, 0.5 $NaHPO_4$, 0.5 $MgCl_2$, 1.5 $CaCl_2$, 15 histidine, and 15 glucose. During anoxia, 5 mM deoxyglucose was substituted for glucose, 35 mM KCl was added to stop spontaneous activity, 50 µM FA-free albumin with 50 µM myristate was added, and solution was degassed by stirring under vacuum.

## Reagents and chemicals

Unless specified otherwise, reagents were from Sigma-Aldrich. NIM811 was a gift of Novartis Pharmaceutical, Basal.

## Statistics

Unless stated otherwise, error bars represent standard error of 6 and usually 8 or more observations. Significance was assessed by Student's t-test or, in rare cases of unequal variance, by the Mann–Whitney Rank Sum test. In all figures, '*' denotes $p < 0.05$, '**' denotes $p < 0.01$, and '***' denotes $p < 0.001$.

## siRNA transfections

Lipofectamine 2000 or RNAiMax transfection reagents were used according to the manufacturer's instructions. Briefly, cells were plated 1 day before transfection in 12-well plates at $2–4 \times 10^5$ cells per well without antibiotics. A non-targeting siRNA (# D-001206-14-05; Thermo Scientific, Waltham, MA) was used as the negative control. A set of three Sigma pre-designed Mission siRNAs were used for ZDHHC5 gene silencing: #1, sense, CGUUACACAGGGUUGCGAAdTdT, anti-sense, UUCGCAACCCUGUGUAACGdTdT; #2, sense, GAUAGUAGCUUAUUGGCCAdTdT, anti-sense, UGGCCAAUAAGCUACUAUCdTdT; #3, sense, GUUUCAGAUGGGCAGAUAAdTdT, anti-sense, UUAUCUGCCCAUCUGAAACdTdT. Control or specific siRNAs were diluted in Opti-MEM (Invitrogen; Grand Island, NY) for 5 min, and Lipofectamine 2000 (Invitrogen) was also diluted in Opti-MEM at room at 23°C for 5 min. Then the diluted siRNA and Lipofectamin 2000 were mixed and incubated for 30 min at 23°C. The siRNA-Lipofectamine 2000 complexes were added to the cells and incubated for 48–72 hr. When employed, STS (0.1 µM; Santa Cruz Biotechechnology, Dallas, TX, # sc-3510) was present throughout the incubation but not during experiments. Effectiveness of siRNA was assessed after 72 hr as demonstrated in *Figure 5A*.

## DHHC2 activity assay

Purification of reagents and methods were as described (*Jennings et al., 2009*). Briefly, CoA solutions were diluted into reactions to the indicated final concentrations of CoA along with 1 µM myrLckNT, 0.85 µM [$^3$H]-palmitoyl-CoA, and 10 nM DHHC2. Reactions were incubated 5 min at 25°C before stopping with 5X sample buffer. Reactions were resolved by SDS-PAGE and either the myrLckNT bands were excised from gels and treated for scintillation counting (*Figure 3B*, above) or gels were treated for fluorography and exposed to film (*Figure 3B*, below). CoA inhibited incorporation of [3H]-palmitate into both DHHC2 and myrLckNT.

## Acknowledgements

We thank Will Fuller (University of Dundee, Dundee), Vincenzo Lariccia (Marche Polytechnic, Ancona), and Beverly Rothermel (UT Southwestern, Dallas) for discussion, Xinlin Du (UT Southwestern, Dallas) for acetyl CoA synthetase, and Hao-Ran Wang (Novartis, Boston) for participation in preliminary studies.

# Additional information

## Funding

| Funder | Grant reference number | Author |
|---|---|---|
| National Institutes of Health | HL513223 | Donald W Hilgemann |
| National Institutes of Health | HL067942 | Donald W Hilgemann |
| National Institutes of Health | GM51466 | Benjamin C Jennings |

The funder had no role in study design, data collection and interpretation, or the decision to submit the work for publication.

## Author contributions

DWH, Conception and design, Acquisition of data, Analysis and interpretation of data, Drafting or revising the article, Contributed unpublished essential data or reagents; MF, M-JL, Conception and design, Acquisition of data, Drafting or revising the article; MEL, Planned DHHC2 palmitoylation assays, Conception and design; Contributed unpublished essential data or reagents; BCJ, Conception and design, Acquisition of data, Analysis and interpretation of data, Contributed unpublished essential data or reagents

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
