## [Decision Letter]

Thank you for sending your work entitled “Massive endocytosis triggered by surface membrane palmitoylation under mitochondrial control in BHK fibroblasts” for consideration at *eLife*. Your article has been favorably evaluated by a Senior editor and 3 reviewers, one of whom is a member of our Board of Reviewing Editors.

The following individuals responsible for the peer review of your submission have agreed to reveal their identity: Richard Aldrich (Reviewing editor) and Bertil Hille (peer reviewer).

The Reviewing editor and the other reviewers discussed their comments before we reached this decision, and the Reviewing editor has assembled the following comments to help you prepare a revised submission.

In this refreshingly original study, Hilgemann and colleagues attempt to understand the mechanism for massive endocytosis (MEND) – uptake of about half the plasma membrane – after a large increase in cytoplasmic calcium. The phenomenon has a corollary in ‘excessive endocytosis’ reported in chromaffin cells by Smith and Neher. The conclusion is that mitochondrial stress induces depolarization or permeability transition pore (PTP) leak of CoA that in turn promote acetyltransferase DHHC5 activity and membrane palmitoylation.

The work brings an innovative mix of biophysical and biochemical techniques and makes excellent use of two techniques unique to this laboratory: perfused whole-cell pipettes and cytoplasmic calcium loading by reversal of the Na/Ca exchanger.

This laboratory discovered MEND and now is making excellent progress towards explaining the signals underlying it, revealing a really new chain of events. The reviewers find the work to be difficult but imaginative, and congratulate the authors on significant discoveries.

Overall the work uses a novel and compelling method of accurate membrane surface area measurement, and intracellular control of metabolites and enzymes, to present a reasonable hypothesis to explain an unusual form of endocytosis. Although the authors tip their hat to the possibility that this pathway is constitutive, the hypothesis mainly rests on dire cellular Ca leak conditions, such as those that occur with membrane damage. This does not make it less important or interesting, and may lead to investigation of new membrane uptake pathways.

While we have high enthusiasm for the paper, the following issues must be resolved before the paper can be accepted.

1) The major problem is that all of the reviewers found the manuscript to be quite difficult to read and to understand. The work is of considerable importance and should be interesting and accessible to a wide range of researchers. We feel that the writing and presentation can be considerably improved so that it will be understandable to all interested readers. We encourage the authors to thoroughly rewrite the paper (and the companion paper) for clarity.

2) The following questions should be addressed, at least by comments in the text. Do PTP channels actually open before MEND occurs? Is MEND actually preceded by a burst of CoA release into the cytosol, and does such release occur through PTP channels? Is MEND preceded by a burst of acyl-CoA synthesis? Does MEND actually require the palmitoylation of a protein, and which? How directly is that protein involved in MEND? Further, how are the findings here related to the requirements for/effects of PIP2, cholesterol, alkylating phosphatases, and sphingomyelinases? In its present form, this paper seems less than definitive.

---

## [Author Response]

*1) The major problem is that all of the reviewers found the manuscript to be quite difficult to read and to understand. The work is of considerable importance and should be interesting and accessible to a wide range of researchers. We feel that the writing and presentation can be considerably improved so that it will be understandable to all interested readers. We encourage the authors to thoroughly rewrite the paper (and the companion paper) for clarity*.

I have rewritten both articles extensively and in doing so I tried to see the article from the perspective of a reader who has never seen an electrophysiological record. The proof of my success is not mine to judge.

*2) The following questions should be addressed, at least by comments in the text. Do PTP channels actually open before MEND occurs? Is MEND actually preceded by a burst of CoA release into the cytosol, and does such release occur through PTP channels? Is MEND preceded by a burst of acyl-CoA synthesis? Does MEND actually require the palmitoylation of a protein, and which? How directly is that protein involved in MEND? Further, how are the findings here related to the requirements for/effects of PIP2, cholesterol, alkylating phosphatases, and sphingomyelinases? In its present form, this paper seems less than definitive*.

I have used these questions almost verbatim to organize the Discussion of the article. I feel that these questions actually bring out valid arguments from the datasets for every important hypothesis raised in the article.